# Controlling Nutritional Status (CONUT) Score is Associated with Overall Survival in Patients with Unresectable Hepatocellular Carcinoma Treated with Lenvatinib: A Multicenter Cohort Study

**DOI:** 10.3390/nu12041076

**Published:** 2020-04-13

**Authors:** Shigeo Shimose, Takumi Kawaguchi, Hideki Iwamoto, Masatoshi Tanaka, Ken Miyazaki, Miki Ono, Takashi Niizeki, Tomotake Shirono, Shusuke Okamura, Masahito Nakano, Hideya Suga, Taizo Yamaguchi, Yoshinori Yokokura, Kazunori Noguchi, Hironori Koga, Takuji Torimura

**Affiliations:** 1Division of Gastroenterology, Department of Medicine, Kurume University School of Medicine, Kurume 830-0011, Japan; shimose_shigeo@med.kurume-u.ac.jp (S.S.); iwamoto_hideki@med.kurume-u.ac.jp (H.I.); niizeki_takashi@kurume-u.ac.jp (T.N.); shirono_tomotake@med.kurume-u.ac.jp (T.S.); okamura_shyuusuke@kurume-u.ac.jp (S.O.); nakano_masahito@kurume-u.ac.jp (M.N.); hirokoga@med.kurume-u.ac.jp (H.K.); tori@med.kurume-u.ac.jp (T.T.); 2Iwamoto Internal Medical Clinic, Kitakyusyu 802-0832, Japan; ttttyama2@yahoo.co.jp; 3Department of Gastroenterology and Hepatology, Yokokura Hospital, Miyama 839-0295, Japan; mazzo6528@me.com (M.T.); yoshinori@yokokura.jp (Y.Y.); 4Department of Gastroenterology and Hepatology, Omuta City Hospital, Omuta 836-8567, Japan; miya3ken@yahoo.co.jp (K.M.); suga516@med.kurume-u.ac.jp (H.S.); 5Department of Gastroenterology and Hepatology, Yanagawa Hospital, Yanagawa 832-0077, Japan; ohno_miki@med.kurume-u.ac.jp (M.O.); hisyo@ghp.omuta.fukuoka.jp (K.N.)

**Keywords:** hepatoma, prognosis, controlling nutritional status, lenvatinib

## Abstract

We aimed to investigate the impact of the controlling nutritional status (CONUT) score, an immuno-nutritional biomarker, on the prognosis of patients with hepatocellular carcinoma (HCC) treated with lenvatinib (LEN). This retrospective study enrolled 164 patients with HCC and treated with LEN (median age 73 years, Barcelona Clinic Liver Cancer (BCLC) stage B/C 93/71). Factors associated with overall survival (OS) were evaluated using multivariate and decision tree analyses. OS was calculated using the Kaplan–Meier method and analyzed using the log–rank test. Independent factors for OS were albumin–bilirubin grade 1, BCLC stage B, and CONUT score <5 (hazard ratio (HR) 2.9, 95% confidence interval (CI) 1.58–5.31, *p* < 0.001). The CONUT score was the most important variable for OS, with OS rates of 70.0% and 29.0% in the low and high CONUT groups, respectively. Additionally, the median survival time was longer in the low CONUT group than in the high CONUT group (median survival time not reached vs. 11.3 months, *p* < 0.001). The CONUT score was the most important prognostic variable, rather than albumin–bilirubin grade and BCLC stage, in patients with HCC treated with LEN. Accordingly, immuno-nutritional status may be an important factor in the management of patients with HCC treated with LEN.

## 1. Introduction

Hepatocellular carcinoma (HCC) is a common malignancy, and one of the leading causes of cancer deaths worldwide [1]. The prognosis of patients with early-stage HCC has improved due to the development of treatment modalities for curative therapy, including radiofrequency ablation [2,3,4]. However, the prognosis of patients with advanced-stage HCC remains unsatisfactory, due to the limited treatment options. In patients with advanced-stage HCC, an oral multikinase inhibitor (MKI) is the only treatment that can improve survival [4,5,6].

Lenvatinib (LEN) is an oral MKI [7] that is approved for first-line treatment of patients with advanced-stage HCC in the United States, European Union, Japan, and China. Kudo et al. reported that LEN is associated with better overall survival (OS) than transarterial chemoembolization (TACE) in patients with large or multinodular intermediate-stage HCC according to the up-to-seven criteria (Child–Pugh class A) [8]. However, the effect of LEN on the OS rate is associated with host factors. The albumin–bilirubin (ALBI) grade is a validated index of liver function in patients with HCC, and it is often used to predict the therapeutic effect on patients treated with LEN [9,10,11].

Recently, immuno-nutritional status has been reported to be associated with outcomes in patients with HCC [12,13,14]. The controlling nutritional status (CONUT) score has been developed as an objective assessment tool of nutritional status [15]. CONUT is scored from 0 to 8 points, and ≥5 points is considered a malnutrition status [15]. The CONUT score is based on three variables: serum albumin level, total cholesterol level, and total lymphocyte count [15]. The CONUT score is now used as an immuno-nutritional index [16,17]. It correlates with disease activity in patients with lupus nephritis [18]. The CONUT score is also associated with poor survival in patients with hepatitis B virus reactivation [19]. In addition, it has been reported to predict the prognosis of patients with various cancers, including gastric cancer, colon cancer, and pancreatic cancer [20,21,22].

The CONUT score has been reported to predict the risk of postoperative complications of grades III–V [23], as well as the prognosis of patients who undergo hepatic resection for HCC [24,25]. Thus, the CONUT score is a useful index for the management of patients who undergo hepatic resection for HCC; however, the impact of a CONUT score on the prognosis for HCC patients treated with LEN has been unclear until now. The aim of this study was to investigate the impact of CONUT score on the prognosis in HCC patients treated with LEN**.**

## 2. Patients and Methods

### 2.1. Study Design

This retrospective study was carried out in five institutions: Kurume University Hospital (Kurume, Japan), Yokokura Hospital (Miyama, Japan), Iwamoto Internal Medical Clinic (Kitakyusyu, Japan), Omuta City Hospital (Omuta, Japan), and Yanagawa Hospital (Yanagawa, Japan). The protocol conforms to the ethical guidelines of the 1975 Declaration of Helsinki, and was approved by the ethics committees of the Kurume University School of Medicine (approval number: 18146), Yokokura Hospital, Omuta City Hospital, Yanagawa Hospital, and Iwamoto Internal Medical Clinic. An opt-out approach was employed to obtain informed consent from the patients, and personal information was protected during data collection.

### 2.2. Inclusion and Exclusion Criteria

The inclusion criteria for this study were as follows: (1) diagnosis of intermediate- or advanced-stage HCC, based on the Barcelona Clinic Liver Cancer (BCLC) staging system [5]; (2) age > 18 years; (3) Eastern Cooperative Oncology Group performance status = 0; and (4) completed follow-ups until death or study cessation (29 February 2020). The exclusion criteria were as follows: (1) history of a malignant tumor, other than HCC, in the five years preceding the study; (2) participation in any clinical trial; (3) Child–Pugh class B or C; (4) creatinine > 1.5 mg/dL; (5) infiltrative HCC; (6) tumor invasion of the portal trunk (Japanese Portal Vein Invasion (VP) staging system class VP 4); (7) presence of ascites; (8) active esophageal varices of form F3; and (9) history of liver transplantation.

### 2.3. Patients

From 24 March 2018 to 28 February 2019, a total of 177 consecutive patients with HCC who received LEN were registered at the Kurume University School of Medicine, Yokokura Hospital, Iwamoto Internal Medical Clinic, Omuta City Hospital, and Yanagawa Hospital. The data cut-off for this analysis was 29 February 2020. Patients meeting any of the exclusion criteria were excluded from the analysis (*n* = 13). A total of 164 patients were enrolled in the study (Appendix A: Figure A1).

### 2.4. Measurement of Biochemical Parameters

We measured biochemical parameters using the standard clinical methods (Department of Clinical Laboratory, Kurume University Hospital): aspartate aminotransferase, alanine aminotransferase, albumin, total bilirubin, total cholesterol, and white blood cells.

### 2.5. Assessment of Immuno-Nutritional Status and Hepatic Functional Reserve

The CONUT scores, used to assess immuno-nutritional status, were calculated from the following three parameters, as previously described [15,26]: (1) albumin levels of 3.5, 3.0–3.49, 2.5–2.99, and <2.5 g/dL were scored as 0, 2, 4, and 6 points, respectively; (2) total lymphocyte counts of ≥1600, 1200–1599, 800–1199, and <800/μL were scored as 0, 1, 2, and 3 points, respectively; (3) total cholesterol levels of ≥180, 140–179, 100–139, and <100 mg/dL were scored as 0, 1, 2, and 3 points, respectively. Separately, the Child–Pugh classification [27] and ALBI grade [28] were used to assess hepatic functional reserve. All patients were classified into the low CONUT group (CONUT score < 5) or the high CONUT group (CONUT score ≥ 5).

### 2.6. Diagnosis of HCC

HCC was diagnosed using a combination of tests for serum tumor markers, such as alpha-fetoprotein (AFP) and des-γ-carboxy prothrombin (DCP), and imaging modalities, such as ultrasonography, computed tomography (CT), and magnetic resonance imaging (MRI). HCC was classified using the BCLC staging system.

### 2.7. Treatment with LEN

After obtaining written informed consent from each patient, treatment with LEN was initiated. The administration dose of LEN is determined based on body weight, according to the manufacturers’ instruction. LEN was orally administered at a dose of 12 mg/day in patients with bodyweight ≥60 kg, or 8 mg/day in patients with bodyweight <60 kg, and it was discontinued when any unacceptable or serious adverse events (AE) were observed.

### 2.8. Follow-Up Schedule and Evaluation of Therapeutic Response

Therapeutic response was evaluated, using CT or MRI, 4–6 weeks after the initiation of treatment with LEN, and thereafter at intervals of 2–3 months until death or study cessation. The therapeutic response was evaluated using the Modified Response Evaluation Criteria in Solid Tumors (mRECIST) [29]. We also evaluated the duration of treatment with LEN at each medical checkup. When HCC recurred, additional treatment was selected based on the evidence-based clinical practice guidelines of BCLC staging and treatment strategy [5].

### 2.9. Safety Evaluation and Assessment of Adverse Events

AE were assessed based on the National Cancer Institute Common Terminology Criteria for Adverse Events, version 4.0 [30]. In accordance with the guidelines for administration of LEN, the dose of LEN was reduced or treatment interrupted when any AE of grade 3 or higher severity or any unacceptable drug-related AE of grade 2 severity occurred.

### 2.10. Clinical Outcomes

The primary endpoint of this study was the OS of the patients.

### 2.11. Statistical Analysis

All data are expressed as the frequency or median (range). All statistical analyses were carried out using statistical analysis software (JMP Pro version 14, SAS Institute Inc., Cary, NC, United States). OS was calculated using the Kaplan–Meier method and analyzed using the log–rank test. Univariate and multivariate analyses were conducted using the Cox proportional hazards model, to identify factors associated with OS. We also performed decision tree analysis to identify factors associated with OS, as previously described [31]. Correlation analysis between CONUT scores and total lymphocyte count, total cholesterol level, and albumin level was performed using simple linear regression analysis. A two-tailed *p*-value of <0.05 was considered statistically significant.

## 3. Results

### 3.1. Patient Characteristics

The patients’ median age was 73 years, and 18.2% of the patients (30/164) were female (Table 1). The median BMI was 22 kg/m^2^. The etiology of HCC was hepatitis C virus in 48% of patients (78/164), and 39% of patients (64/164) had an ALBI grade of 1. The median total cholesterol level and lymphocyte count were 171 mg/dL and 4600/μL, respectively. The frequencies of CONUT scores of 0–1 (normal nutrition), 2–4 (mild malnutrition), 5–8 (moderate malnutrition), and ≥9 (severe malnutrition) were 29.3% (48/164), 49.3% (81/164), 20.8% (34/164), and 0.6% (1/164), respectively (Table 1). The median tumor size was 32.5 mm, and 76% of patients (93/164) were classified as Barcelona Clinic Liver Cancer (BCLC) stage B (Table 1).

### 3.2. Evaluation with mRECIST After Treatment with LEN

One month after treatment, a complete response (CR), partial response (PR), stable disease (SD), and progressive disease (PD) were observed in 3% (5/164), 30% (49/164), 41% (68/164), and 26% (42/164) of patients, respectively (Table 2). The overall objective response rate (ORR) and disease control rate (DCR) were 33% (54/164) and 74% (122/164), respectively (Table 2).

### 3.3. Univariate and Multivariate Analyses of Factors Potentially Associated with Overall Survival

Using univariate analysis, ALBI grade 1, CONUT score <5, BCLC stage B, and AFP ≤200 ng/mL were selected as variables. Using multivariate analysis, ALBI grade 1, CONUT score <5, BCLC stage B, and AFP ≤200 ng/mL were identified as independent factors for OS (Table 3).

### 3.4. Kaplan–Meier Survival Analysis for All Patients Treated with LEN

Kaplan–Meier survival analysis for all patients treated with LEN is shown in Figure 1. The median survival time (MST) was 17.6 months. The OS rate was 67% at 1 year.

### 3.5. Decision Tree Survival Analysis

In this study, the survival rate in all subjects was 61% at the time of study cessation. To determine the profiles associated with survival rates, decision tree analysis was performed, which revealed that the CONUT score was the splitting variable for survival rate. Although the survival rate was 29% in patients with CONUT scores ≥5, the survival rate was 70% in patients with CONUT scores <5 (Figure 2). In patients with CONUT scores <5, the second and third splitting variables were BCLC stage and ALBI grade, respectively. The survival rate was 90% in patients with CONUT scores <5, BCLC stage B, and ALBI grade 1 (Profile 1 in Figure 2). In contrast, the survival rate was 8% in patients with CONUT scores ≥5 and BCLC stage C (Profile 5 in Figure 2).

### 3.6. Kaplan–Meier Survival Analysis According to CONUT Score

Kaplan–Meier survival analysis according to CONUT score is shown in Figure 3. All patients were classified into the low CONUT group (CONUT score <5) or the high CONUT group (CONUT score ≥5). The survival rate of the low CONUT group was significantly higher than that of the high CONUT group. In the low CONUT group, the 3-, 6-, and 12-month survival rates were 97%, 89%, and 76%, respectively. In the high CONUT group, the 3-, 6-, and 12-month survival rates were 90%, 76%, and 43%, respectively (Figure 3).

### 3.7. Correlations Between CONUT Score and Total Lymphocyte Count, Total Cholesterol level, and Albumin Level

There was a significant negative correlation between CONUT scores and total lymphocyte count (Figure 4a). Significant negative correlations were also seen between CONUT scores and total cholesterol level and albumin levels (Figure 4b,c).

### 3.8. Duration of Treatment with LEN According to CONUT Score

The duration of treatment with LEN according to CONUT score is shown in Appendix A, Figure A2. Treatment duration was significantly longer in the low CONUT group (CONUT score <5) than in the high CONUT group (CONUT score ≥5). Median treatment times were 9.7 months and 3.5 months in the low CONUT score group and the high CONUT score group, respectively.

### 3.9. Cessation of Treatment with LEN Due to Severe Adverse Events

Cessation of treatment with LEN due to severe AE occurred in 45% (74/164) of all patients. Cessation of treatment was significantly more frequent in the high CONUT group than in the low CONUT group (39% in the low CONUT group vs. 69% in the high CONUT group; *p* = 0.001). We examined the impact of the parameters of the CONUT on discontinuation of LEN due to AE. There was no significant difference in total lymphocyte count and total cholesterol level between the discontinuation and no discontinuation due to AE groups (Figure A3). However, the serum albumin levels were significantly higher in the no discontinuation due to AE group than in the discontinuation due to AE group (Figure A3).

## 4. Discussion

Through this study, we demonstrated that a CONUT score <5, ALBI grade 1, BCLC stage B, and AFP ≤200 ng/mL were independently associated with OS in patients with HCC treated with LEN. Moreover, through decision tree analysis, we revealed that the CONUT score was the initial splitting variable for survival rate. These results indicate that, alongside liver function and tumor factors, immuno-nutritional status may be an important factor in the management of patients with HCC treatment with LEN.

Treatment with LEN is a recommended therapy for unresectable HCC, according to the clinical guidelines for HCC in the United States, Europe, and Japan [3,5,6,32]. In our study, the MST was 17.6 months in patients with HCC treated with LEN. Furthermore, in our study, the ORR and DCR were 33% and 74%, respectively. Yamashita et al. previously reported an MST of 17.6 months in Japanese patients with unresectable HCC treated with LEN in a phase 3, multinational, randomized, non-inferiority trial (REFLECT) [33]. Thus, the therapeutic effect of LEN in our study appears to be similar to that reported previously [7,11,33], suggesting that the enrolled subjects and treatment effects in our study are typical.

In this study, a CONUT score <5, ALBI grade 1, BCLC stage B, and AFP ≤200 ng/mL were identified as independent factors for OS, using multivariate analysis. Previous studies have reported that ALBI grade, AFP level, and BCLC stage are independent factors for OS in patients with HCC treated with LEN [7,9], indicating that our results showed good agreement with those of previous reports. The CONUT score has previously been reported as a prognostic factor for patients with HCC treated with hepatic resection [24,25,34]. However, we are the first to reveal that the CONUT score is an independent factor for OS in patients with HCC treated with LEN. This indicates that the CONUT score is a prognostic factor independent from liver function and tumor factors. Various factors are involved in the prognoses of patients with HCC treated with LEN; however, the prognostic importance of each factor has been unclear until now.

In this study, we investigated the prognostic importance of each factor for OS, using decision tree analysis. We found that the CONUT score was the initial splitting variable for survival rate in patients with HCC treated with LEN, followed by BCLC stage and ALBI grade. Thus, the CONUT score may be the most important prognostic factor in patients treated with LEN. Daitoku et al. report that the CONUT score is a prognostic factor in patients with metastatic colorectal cancer treated with chemotherapy [35]. In fact, the OS rate of the low CONUT score group was significantly higher than that of the high CONUT score group in our study. It is unclear why the CONUT score was the most important predictive factor for OS in patients with HCC treated with LEN. In fact, we performed the decision tree analysis with three parameters comprising the CONUT score separately. In the analysis, the BCLC stage was identified as the variable for the initial split of survival (data not shown). However, none of three parameters comprising the CONUT score was identified as the initial split of survival, suggesting that the combined power of the CONUT score was important for overall survival in patients treated with LEN.

The CONUT score is based on total lymphocyte count, total cholesterol level, and serum albumin level. All three of these factors showed significant correlations with total CONUT score in this study. All three are also well-known nutritional parameters. Protein energy malnutrition has been reported to downregulate protein/hormone and antibody synthesis [36,37], leading to suppression of both B-lymphocyte differentiation [38] and T-lymphocyte regulation [39]. In fact, the total lymphocyte count is correlated with anthropometric nutritional indicators, such as mid-upper arm circumference and triceps skinfold thickness [40]. In addition, the total lymphocyte count is utilized as an immune marker, and is known to correlate with the CD4+ lymphocyte count [41,42]. Moreover, cholesterol localizes in lipid rafts of the cell membrane. The depletion of cholesterol in the lipid rafts is reported to be a trigger for ligand-independent activation of the epidermal growth factor receptor [43]. Furthermore, albumin is the most abundant protein in plasma, and is known to act as an antioxidant via radical scavenging [44]. This suggests that the CONUT score may reflect not only nutritional status, but also immunoresponsiveness, cell membrane lipid raft status, and antioxidative capacity. This may explain why the CONUT score has been identified in this study as the most important variable associated with prognosis in patients with HCC treated with LEN.

The cessation of treatment due to severe AE was significantly more frequent in the low CONUT group than in the high CONUT group in our study. This suggests that a lower CONUT score may be associated with better OS through the longer duration of treatment with LEN. We examined the impact of the parameters of the CONUT on discontinuation of LEN due to AE. We found that serum albumin level, but not total lymphocyte count or total cholesterol level, was significantly higher in the no discontinuation due to AE group than in the discontinuation due to AE group. Since lenvatinib is known to mainly bind to human serum albumin [45], a higher serum albumin level may contribute to the suppression of the development of AE.

This study could inform the experimental design of future prospective trials in this clinical context. Our results may indicate the importance of evaluation for nutritional state of patients, using CONUT as a base line and throughout the study. In addition, cholesterol is a precursor for bile acids and various other bioactive molecules, such as oxysterols [46], which could have an impact on LEN uptake and action. Thus, deeper probing of other metabolic parameters, including bile acids and immune status, is also required in future prospective trials.

This study has several limitations. First, the study was retrospective. Second, we did not evaluate patient histories of previous and subsequent treatment for HCC. Third, we did not evaluate sarcopenia, which has a significant relationship with both nutrition and prognosis. To prove a prognostic impact of the CONUT score in patients with HCC treated with LEN, it would be necessary to perform a prospective study, including analysis of various host and tumor factors.

In this study, we showed that a CONUT score <5, ALBI grade 1, BCLC stage B, and AFP ≤200 ng/mL were independently associated with better prognosis in patients with HCC treatment with LEN. Moreover, we revealed that the CONUT score was the most important variable for OS. Accordingly, immuno-nutritional status should be an important factor in the management of patients with HCC treatment with LEN. Future work will focus on nutritional therapy for the improvement of CONUT scores in patients with HCC treated with LEN.

## Figures and Tables

**Figure 1 nutrients-12-01076-f001:**
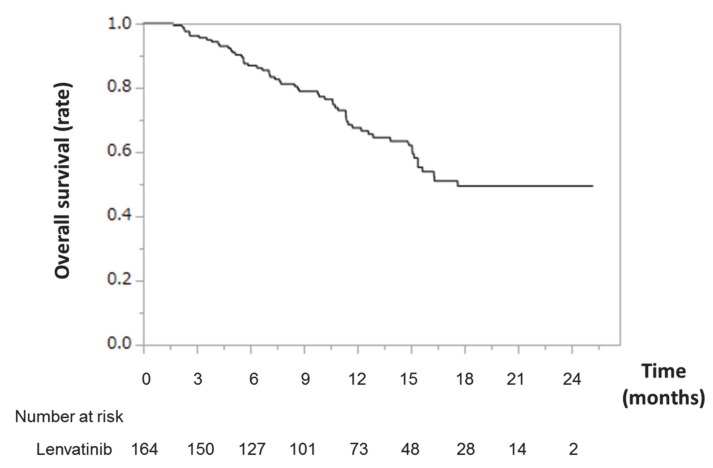
Overall survival time in patients with HCC treated with lenvatinib. Kaplan–Meier curves for overall survival time in patients with HCC treated with lenvatinib. Abbreviations: HCC, hepatocellular carcinoma.

**Figure 2 nutrients-12-01076-f002:**
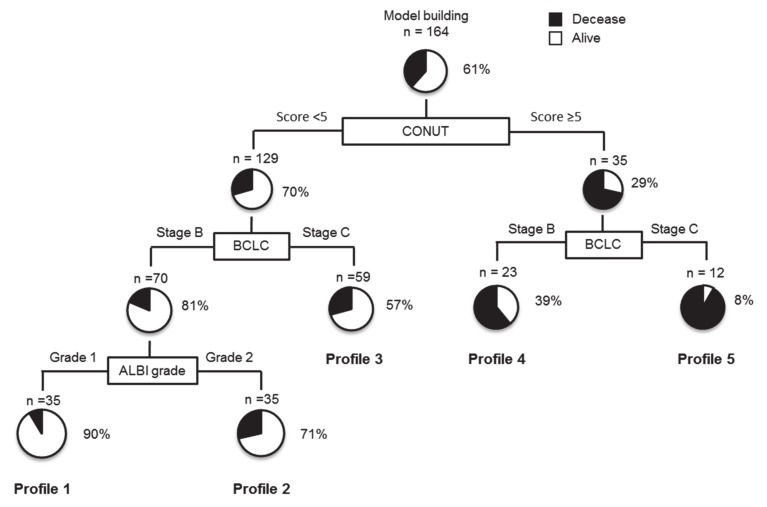
Profiles associated with survival in patients with HCC treated with lenvatinib. Decision tree algorithm for OS. Pie graphs indicate the percentages of alive (white)/deceased (black) patients in each group. Abbreviations: HCC, hepatocellular carcinoma; OS, overall survival.

**Figure 3 nutrients-12-01076-f003:**
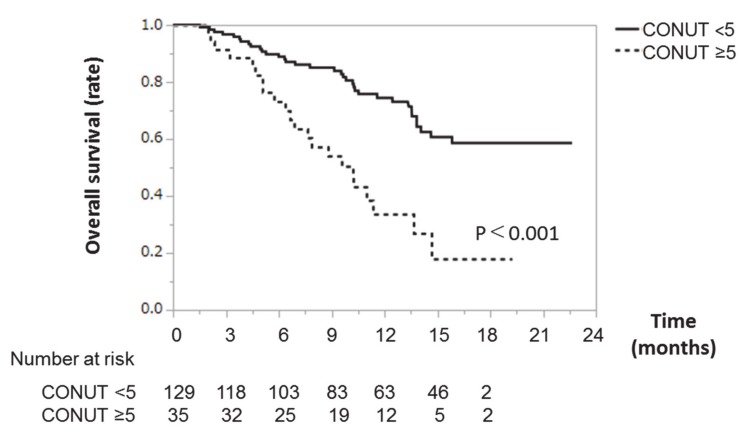
Overall survival time in patients with HCC treated with lenvatinib. Kaplan–Meier survival analysis, showing overall survival time according to CONUT score (<5 or ≥5) in patients with HCC treated with lenvatinib. The solid line represents the low CONUT group (CONUT score <5). The dotted line represents the high CONUT group (CONUT score ≥5). Abbreviations: HCC, hepatocellular carcinoma; CONUT, Controlling Nutritional Status.

**Figure 4 nutrients-12-01076-f004:**
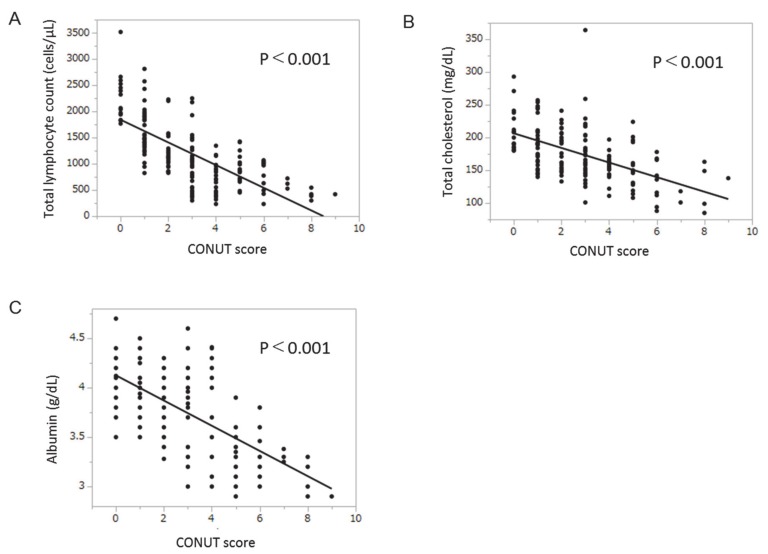
Correlations between CONUT score and total lymphocyte count, total cholesterol level, and albumin level. (**a**) Correlation between CONUT score and total lymphocyte count; (**b**) correlation between CONUT score and total cholesterol level; (**c**) correlation between CONUT score and albumin level. Abbreviations: CONUT, Controlling Nutritional Status.

**Table 1 nutrients-12-01076-t001:** Patient characteristics.

Characteristic	Patients (*n* = 164)
Age (years)	73 (42–89)
Sex (female/male)	30/134
BMI (kg/m^2^)	22 (15–38.9)
Cause of HCC (HBV/HCV/Other)	32/78/54
AST (U/L)	33 (13–160)
ALT (U/L)	30 (6–120)
Albumin (g/dL)	3.8 (2.9–4.7)
Total bilirubin (mg/dL)	0.8 (0.2–2.4)
Child–Pugh score (A/B)	158/6
ALBI grade (1/2)	64/100
Diabetes mellitus (+/-)	68/96
Total cholesterol (mg/dL)	171(85–364)
CONUT score 0–1 (normal nutrition) 2–4 (mild malnutrition) 5–8 (moderate malnutrition) ≥9 (severe malnutrition)	48 81 34 1
Maximum tumor diameter (mm)	32.5 (10–127)
Number of tumors <5/≥5	43/121
BCLC stage (B/C)	93/71
AFP (ng/mL)	51.2 (1.0–146,260)
DCP (mAU/mL)	233.5 (3.3–524,068)

Data are expressed as median (range), or frequency. Abbreviations: BMI, body mass index; HBV, hepatitis B virus; HCV, hepatitis C virus; ALBI, albumin–bilirubin; CONUT, Controlling Nutritional Status; BCLC, Barcelona Clinic Liver Cancer; AFP, α-fetoprotein; DCP, des-γ-carboxy prothrombin.

**Table 2 nutrients-12-01076-t002:** Treatment response rate.

Response Category	Patients with HCC Treated with Lenvatinib (*n* = 164)
CR	5 (3%)
PR	49 (30%)
SD	68 (41%)
PD	42 (26%)
ORR	54 (33%)
DCR	122 (74%)

Data are expressed as frequency (percentage). Abbreviations: HCC, hepatocellular carcinoma; CR, complete response; PR, partial response; SD, stable disease; PD, progressive disease; ORR, objective response rate; DCR, disease control rate.

**Table 3 nutrients-12-01076-t003:** Univariate and multivariate analyses of factors potentially associated with OS.

Variable	Univariate Analysis	Multivariate Analysis		
	*p-*Value	HR	95% CI	*p-*Value
Age (<65 years/≥65 years)	0.779			
Gender (female/male)	0.544			
BMI (<22 kg/m^2^/≥22 kg/m^2^)	0.199			
HCC etiology (HBV/HCV/Others)	0.634			
ALBI grade (2/1)	<0.001	2.446	(1.277–5.202)	0.01
CONUT score (≥5/<5)	<0.001	2.911	(1.579–5.351)	<0.001
BCLC stage (C/B)	0.005	2.771	(1.611–4.844)	<0.001
AFP (≥200 ng/mL/<200 ng/mL)	0.01	1.745	(1.024–2.948)	0.04
DCP (≥200 mAU/mL/<200 mAU/mL)	0.06			

Abbreviations: OS, overall survival; HR, hazard ratio; CI, confidence interval; BMI, body mass index; HCC, hepatocellular carcinoma; HBV, hepatitis B virus; HCV, hepatis C virus; ALBI grade, albumin-bilirubin grade; CONUT, Controlling Nutritional Status; BCLC, Barcelona Clinic Liver Cancer; AFP, α-fetoprotein; DCP, des-γ-carboxy prothrombin.

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
