# Peer review of "Controlling Nutritional Status (CONUT) Score is Associated with Overall Survival in Patients with Unresectable Hepatocellular Carcinoma Treated with Lenvatinib: A Multicenter Cohort Study"

_nutrients, 2020, doi:10.3390/nu12041076_

Round 1

Reviewer 1 Report

Shimose et al. describe results of a retrospective, observational study in LEN treated HCC patients, evaluating the prognostic power of the CONUT score in this cohort of 164 patients. Compared to a number of other evaluated variables, the CONUT score was identified as the factor with the highest predictive power with OS as primary end point. Their study underlines the importance of taking nutritional status into account during cancer treatment protocols. In particular, this is true for liver cancer as the liver is a primary response organ to nutrient composition and fluctuations. The manuscript is written in a concise way and the data analyses are done with methods appropriate for such a study. Therefore, this reviewer only has minor concerns:

-) Can the authors comment on why the liver function markers ALT and AST were not evaluated in this study?

-) For the purpose of this manuscript, a more detailed description on how the decision tree analysis was performed would be helpful. For instance, in their cohort, BCLC stage has a similar high hazard ratio than CONUT score. Could the decision tree analysis be performed with BCLC stage as first separation variable to compare the results to the analysis in Fig 2, or is this an entirely unsupervised analysis?

-) Correlation of low CONUT score with OS seems to be mainly driven by longer duration of treatment. This is a key finding. Hence, the authors should elaborate more on the discussion of which parameters of the CONUT score could be connected to adverse effects during LEN treatment. For instance, cholesterol is not only important in lipid rafts, but is also a precursor for bile acids (among many other bioactive molecules such as oxysterols) that could have an impact on LEN uptake and action.

-) 264-265: How is lymphocyte count a nutritional parameter?

-) This reviewer would appreciate more discussion about how this study could inform the experimental design of future prospective trials in this clinical context (e.g. evaluating nutritional state and metabolic conditions of patients at base line and throughout the study, deeper probing of other metabolic parameters (e.g. bile acids, see above) and the immune status (CRP and other inflammation markers)).

-) To show the combinatorial power of the CONUT score, it would be helpful to perform the decision tree analysis with each of the 3 parameters comprising the CONUT score separately, if only to show that no single parameter harbours the predictive value.

Reviewer 2 Report

Review:

The aim of this manuscript is to to investigate the impact of CONUT score on  prognosis in HCC patients treated with LEN.

The aim is correct because other data indicate that CONUT score correlates with disease activity in patients with lupus nephritis. CONUT score is also associated with poor survival in patients with hepatitis B virus reactivation. In addition, CONUT score has been reported to predict the prognosis of patients with various cancers. 

The methods is well choosen.

The conclusion not spectaculary. Future study are necesarry.

The minor revision I suggest

The authors should add information  about the methods used to detection of serum markers.

Moreover the authors should explain the time of conducted studies.

The authors should discuss about the association with doses LEN 12mg/kg or 8mg/kg and results.
